# A human subcortical network underlying social avoidance revealed by risky economic choices

Johannes Schultz[1,2,3]*, Tom Willems[1], Maria Gädeke[1], Ghada Chakkour[1,4], Alexander Franke[1,4], Bernd Weber[2,3], Rene Hurlemann[1,5]

[1]Division of Medical Psychology, University of Bonn, Bonn, Germany; [2]Center for Economics and Neuroscience, University of Bonn, Bonn, Germany; [3]Institute of Experimental Epileptology and Cognition Research, University of Bonn, Bonn, Germany; [4]Medical School, University of Bonn, Bonn, Germany; [5]Department of Psychiatry and Psychotherapy, University of Bonn, Bonn, Germany

**Abstract** Social interactions have a major impact on well-being. While many individuals actively seek social situations, others avoid them, at great cost to their private and professional life. The neural mechanisms underlying individual differences in social approach or avoidance tendencies are poorly understood. Here we estimated people's subjective value of engaging in a social situation. In each trial, more or less socially anxious participants chose between an interaction with a human partner providing social feedback and a monetary amount. With increasing social anxiety, the subjective value of social engagement decreased; amygdala BOLD response during decision-making and when experiencing social feedback increased; ventral striatum BOLD response to positive social feedback decreased; and connectivity between these regions during decision-making increased. Amygdala response was negatively related to the subjective value of social engagement. These findings suggest a relation between trait social anxiety/social avoidance and activity in a subcortical network during social decision-making.
DOI: https://doi.org/10.7554/eLife.45249.001

*For correspondence:
johannes.schultz@ukbonn.de

**Competing interests:** The authors declare that no competing interests exist.

## Introduction

Pursuing interpersonal relationships is a powerful human drive. Social relationships contribute to the feeling that life has meaning (*Baumeister and Leary, 1995*) and social isolation is a major health risk factor (*House et al., 1988*; *Cacioppo et al., 2015*) with an influence on mortality risk comparable with smoking or alcohol consumption (*Holt-Lunstad et al., 2010*). Several social stimuli and situations can act as a reward (*Krach et al., 2010*), including beautiful faces (*Aharon et al., 2001*; *O'Doherty et al., 2003*), praise and attention (*Izuma et al., 2008*), anticipation of positive social feedback (*Spreckelmeyer et al., 2009*) and interactions in the game Cyberball (*Kawamichi et al., 2016*). However, the value of engaging in social interactions varies across individuals (*Cheek and Buss, 1981*). Social situations are a source of pleasure for sociable people for which they invest time and money, such as when inviting friends for celebrating one's birthday (*Dunn et al., 2008*). In contrast, socially anxious people tend to avoid social situations in order to avoid receiving negative social feedback and experiencing feelings such as fear and shame (*Clark and Wells, 1995*).

The neural mechanisms underlying the decision to engage in a social situation and their variations across individuals are unclear. To better understand these mechanisms, we first aimed to quantify how much individuals value engaging in a simple social situation. We measured the choice frequency of engaging in the social situation as opposed to obtaining a monetary amount, from which we inferred the value that engagement has for a given individual. A reduced frequency of choosing the

**eLife digest** Your relationships with the people around you – friends, family, colleagues – have a strong influence on your overall life happiness. Even so, many people struggle to engage with the people around them. Social interactions can be stressful and many people choose to avoid them, even at a cost.

Being able to measure these tendencies experimentally is a first useful step for assessing social avoidance without relying on people's, often biased, recollections of their actions and behaviours. But how can a tendency to avoid social situations be quantified? And what can an experiment to measure this tendency reveal about the neural underpinnings of social avoidance?

Schultz et al. asked volunteers to play a social game. If they played, the volunteers had the chance to win three euros, but they could choose not to play and receive a fixed amount of money, which varied across trials between zero and three euros. This approach allowed Schultz et al. to quantify how much the volunteers valued playing the game. The game involved playing with other virtual human partners, who gave either positive or negative social feedback depending on the outcome of the game in the form of videos of facial expressions. In a non-social control experiment, a computer gave abstract feedback in the form of symbols.

Schultz et al. found that the value people placed on playing the social game varied with their level of social anxiety (established using a standard questionnaire). The more anxious people attributed less value to engaging in the game. Neuroimaging experiments revealed that the activity and connectivity between the amygdala and ventral striatum, two parts of the brain involved in processing emotions and reward-related stimuli, varied according to people's levels of social anxiety.

Social interactions have a major impact on the quality of life of both healthy people and those with mental disorders. Developing new ways to measure and understand the differences in the brain linked to social traits could help to characterise certain conditions and document therapy progress. Methods to quantify social anxiety and avoidance are also in line with efforts to explore the neuroscience behind the full range of human behaviour.
DOI: https://doi.org/10.7554/eLife.45249.002

social situation would indicate a reduced value of the situation, and a tendency to avoid the social situation. We hypothesised that considering whether to engage in a social situation and experiencing its outcome would evoke avoidance-related neural signals proportional to the individual level of social anxiety, specifically increased BOLD response in the amygdala, a neural structure strongly associated with fear and avoidance (*Shackman and Fox, 2016*; *Terburg et al., 2018*), decision-making (*Jenison et al., 2011*; *Grabenhorst et al., 2012*) and stimulus valuation (*Paton et al., 2006*). Further, we hypothesised that the reduced value of the social situation associated with higher levels of social anxiety would be reflected in reduced reward-related activations in the ventral striatum (*Schultz et al., 1997*; *O'Doherty et al., 2004*) in response to positive outcomes of the situation. Finally, we hypothesised that social anxiety may affect functional connectivity between amygdala and ventral striatum, a pathway implicated in active avoidance behaviour (*Ramirez et al., 2015*).

## Results

We conducted two studies on separate groups of healthy participants. We quantified social anxiety and related personality traits used to assess the specificity of our results with standard questionnaires. In Study 1, we evaluated whether the value of engaging in a social situation varied with a widely accepted measure of trait social anxiety, the Liebowitz Social Anxiety Score, LSAS (*Rytwinski et al., 2009*). Because high autistic traits are associated with deficits in social communication (*American Psychiatric Association, 2013*) and social anxiety (*Kuusikko et al., 2008*) yet described by partly distinct diagnostic criteria from social anxiety, we included autism quotient [AQ (*Baron-Cohen et al., 2001*)] as a covariate. Further, we included participant gender as predictor because gender is known to influence social anxiety (*Xu et al., 2012*). In Study two we replicated and extended these findings by evaluating effects of LSAS, AQ, gender, as well as two scores measuring traits frequently associated with social anxiety (*Acarturk et al., 2008*), namely depression

[Beck's Depression Index, BDI (*Beck et al., 1996*)] and trait general anxiety [STAI-T (*Spielberger and Gorsuch, 1970*)]. Descriptive statistics, internal consistency and reliability measures of these questionnaire data are provided in *Supplementary file 1A and 1B*. We then investigated the neural mechanisms involved in our task using a slightly modified version of the task.

On every trial (*Figure 1A–B*), participants chose between a risky option consisting of a monetary gamble with two potential known outcomes (0 and 3 Euros), presented as a game of dice against a virtual partner (one of six human characters, *Figure 1C*, or a computer as control) and a safe option consisting of a certain amount of money. Critically, choosing the risky option exposed participants to a display of the partner's reaction to the game outcome (*Figure 1D*). The human partner's reaction consisted of a video displaying a facial expression of admiration (i.e., positive social feedback) or condescension (i.e., negative social feedback) depending on whether the participant won or did not win (respectively), while the computer's reactions were abstract symbols. We measured the frequency of choosing the risky or the safe option for different monetary amounts offered in the safe option. These choice frequencies allowed us to quantify the subjective value of engaging in the game against both partners and thereby estimate participants' social avoidance or approach tendencies.

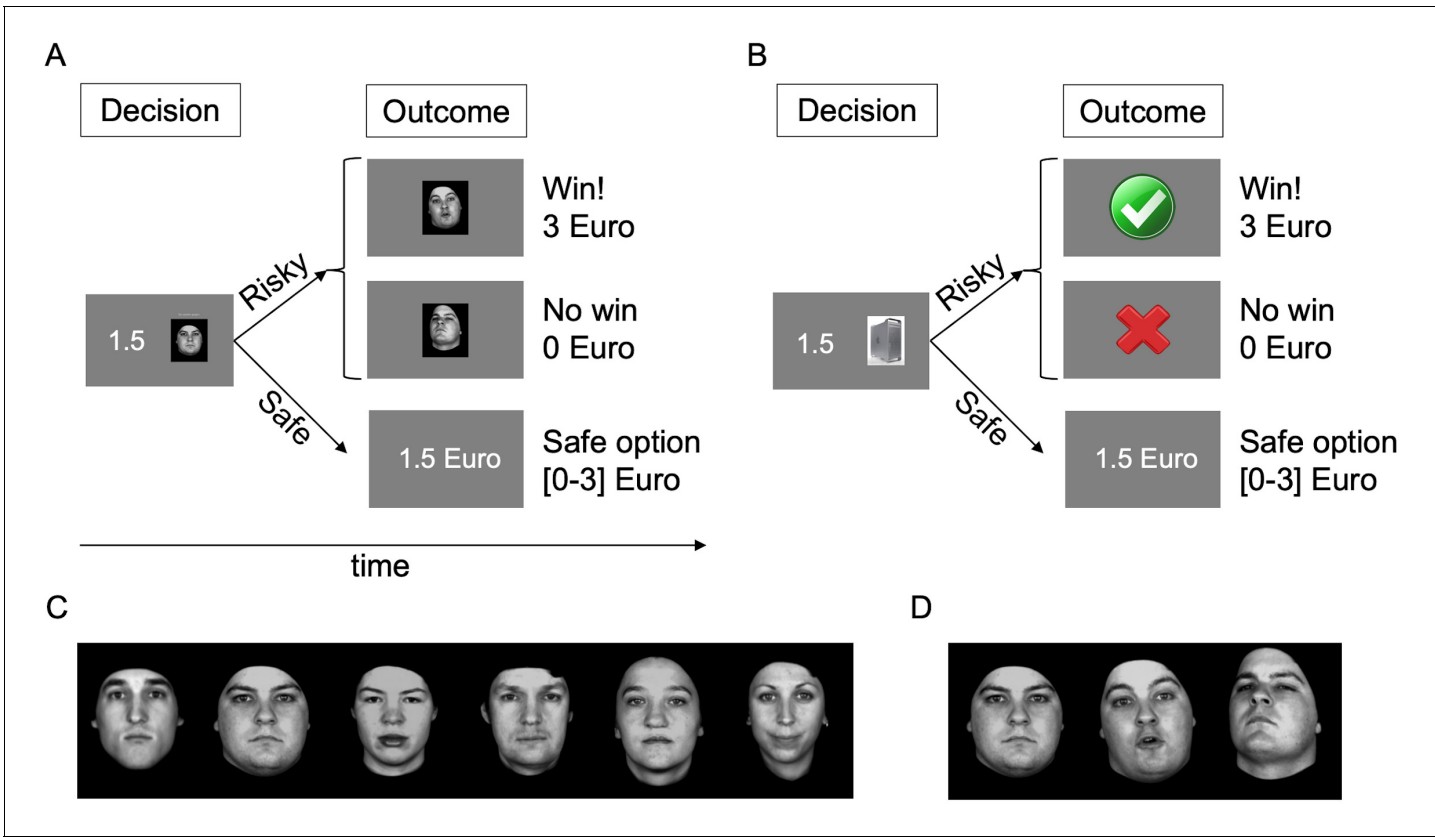

**Figure 1.** Task. (A) Trial structure with human partner. Participants chose between a risky and a safe option. Risky options were monetary gambles, in the form of a virtual game of dice played against a partner presented as an image, with two known potential outcomes (0 or 3 Euros). In the fMRI task, these images were replaced with the name of the partner, learned before the scan. Choosing the safe option led to the receipt of a certain monetary amount, which was varied across trials (0 to 3 Euros, 0.5 Euro steps, equal probability, random order). Choosing the risky option resulted in the display of the outcome of the gamble (win, 3 Euro or no win, 0 Euro) and the display of the partner's response. At the end of the game, the amount gained in one randomly selected trial was paid out in real currency. (B) Trial structure with computer partner, identical to trial with human partner except for the fact that the partner was a computer. (C) The six human partners with neutral facial expression. (D) One of the partners' faces displaying no emotion, admiration and condescension (left to right); the emotional expressions were presented as videos.
DOI: https://doi.org/10.7554/eLife.45249.003

## Behaviour (study 1)

As expected, participants' likelihood of choosing the safe option and thus avoid the gamble increased as a function of the amount offered as an alternative to the gamble, with both human and computer partners (*Figure 2A*). Based on the choice frequencies, we calculated the certainty equivalent ($CE_{50}$) of the gamble against each partner ($CE_{50\ Human}$; $CE_{50\ Computer}$), which represents a measure of the subjective value of engaging in the situation. We expected that the value of engaging in the gamble against the human, but not against the computer, would be lower in participants with more pronounced social anxiety traits (=higher LSAS score). We thus computed the difference between the estimated values of the two gambles ($CE_{50\ Human}$ minus $CE_{50\ Computer}$), which yields a measure of the value of social engagement controlled for variations in risk preference. A multiple regression model with LSAS, AQ and gender as predictor variables could explain individual differences in the value of social engagement (global $F(3,64) = 3.2$, p=0.028, $R^2$=0.13). Only LSAS contributed significantly to the prediction of the interindividual variations in the value of social engagement (value decreased with LSAS: $t(64) = -2.99$, p=0.004, $P_{corr} = 0.012$ Bonferroni-corrected for the use of three predictor variables, unstandardized coefficient = $-0.01$ Euro/LSAS point, adjusted $R^2 = 0.09$, *Figure 2C*, *Supplementary file 1C*), while variations in AQ and gender did not (respectively: $t(64) = 1.75$, p=0.084; $t(64) = 0.24$, p=0.81).

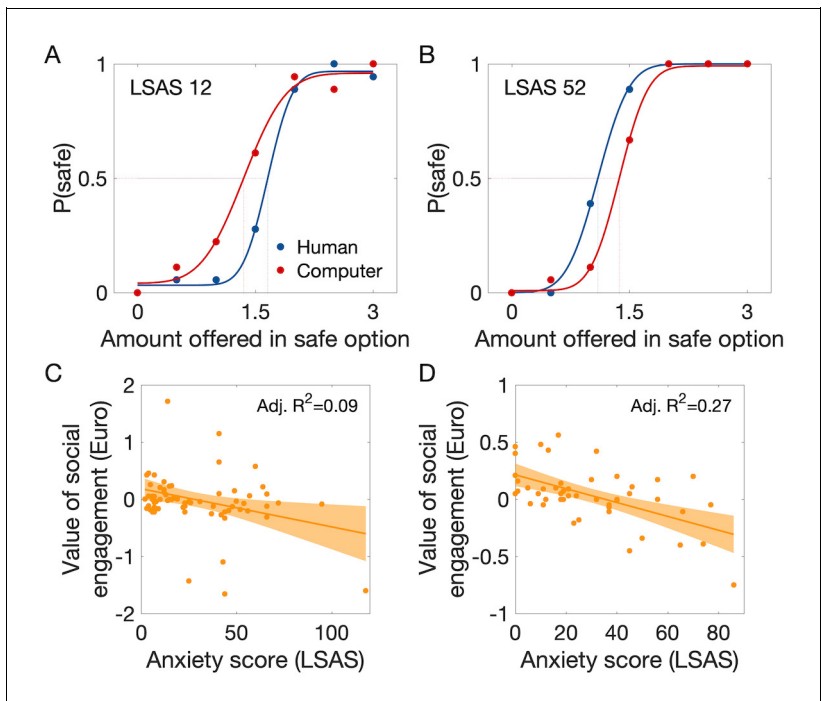

**Figure 2.** Behavioural results showing an association between the value of social engagement and social anxiety level. (**A**) Probability of choosing the safe (no gamble) option instead of the gamble as a function of the amount offered in the safe option, with fitted cumulative Gaussian functions (red and blue lines). The value of engaging in each kind of gamble was estimated as the certainty equivalent of the gamble ($CE_{50}$), which is the amount offered in the safe option associated with 50% safe choices. Example data are shown from participants with anxiety score (LSAS) below (**A**) and above the median (**B**): less anxious participants tended to choose the safe option against the computer partner at lower amounts than against the human partner, indicating a tendency to seek the social situation; more anxious participants showed the opposite tendency, indicating social avoidance. Each marker is the mean of 10 trials. (**C**) The value of social engagement ($CE_{50\ Human}$ – $CE_{50\ Computer}$) decreased with social anxiety score (LSAS) (Study 1; n = 68). (**D**) Replication of the findings in Study 2 (n = 47); same display as in B. P (safe)=probability of choosing the 'safe' option [0–1]. Grey-shaded areas represent the 95% confidence intervals around the slope of the regression line.
DOI: https://doi.org/10.7554/eLife.45249.004

To test if differences in avoidance as a function of social anxiety were due to differences in the perception of the social feedback, we analysed valence ratings of the feedback stimuli. As expected, ratings of positive and negative feedback differed ($F(1,66) = 355.8$, p<0.001, $\eta_p^2=0.84$, higher ratings for positive feedback), ratings for human and computer partners did not ($F(1,66) = 0.4$, p=0.43, $\eta_p^2=0.005$), and ratings differed more between negative and positive human feedback than computer feedback (interaction between partner and positive vs. negative feedback: $F(1,66) = 234.8$, p<0.001, $\eta_p^2=0.78$). Importantly, none of these ratings varied with social anxiety (all $F(1,66) < 0.24$, all p>0.6, all $R^2<0.01$).

## Behaviour (study 2, pre-fMRI)

We replicated our findings in a separate group of 47 participants and verified the specificity of the effects of social anxiety by including more personality trait measures. A multiple linear regression with LSAS, AQ, gender, BDI and STAI - T as predictor variables could explain individual differences in the value of social engagement (global $F(5,39) = 4.3$, p=0.004, $R^2=0.35$). Again, only LSAS contributed to predicting interindividual variations in the value of social engagement (value decreased with LSAS: $t(39) = -3.4$, p=0.002, $P_{corr} = 0.01$ Bonferroni-corrected for the use of five predictor variables, unstandardized coefficient $= -0.007$ Euro/LSAS point, adjusted $R^2 = 0.27$, *Figure 2D*, *Supplementary file 1D*), while variations in AQ, gender, BDI and STAI-T did not (all $|t(39)|<1.4$, p>0.17). Individual values of engaging in the gamble against human and computer partners ($CE_{20}$, $CE_{50}$ and $CE_{80}$) obtained in this behavioural experiment were used during scanning of these participants in a slightly modified version of the task (Materials and methods).

## Behaviour (study 2, during fMRI)

Participants underwent fMRI scans after the behavioural experiment; the data of 42 participants were analysed (data from five participants with excessive head motion were excluded; see Materials and methods). To increase comparability of the BOLD data across participants, we aimed to equalise the ratio of safe vs. risky choices taken by each participant during scanning by using individually calculated $CE_{20}$, $CE_{50}$ and $CE_{80}$ for the gamble against human and computer partners (see above and Materials and methods). Data acquired during scanning revealed that, as expected, the proportion of safe choices increased from $CE_{20}$ through $CE_{50}$ to $CE_{80}$ ($F(2,80) = 53.9$, p<0.001, $\eta_p^2=0.57$), that the proportion of safe choices did not significantly vary between human and computer partners ($F(1,40) = 2.35$, p=0.13, $\eta_p^2=0.06$), that there was no significant interaction between these factors ($F(2,80) = 1.96$, p=0.15, $\eta_p^2=0.05$), and crucially that there were no effects of social anxiety on these decisions (all $F(1,40) < 2.3$, all p>0.1, all $\eta_p^2<0.06$). Two participants who never chose the risky option against the human partner during the scan were excluded from further analysis because their neural response during that crucial experimental condition could not be assessed.

## Estimating a cost of social avoidance

Based on the regression results linking social anxiety to variations in the subjective value of engaging in the social situation, we estimated a cost of social avoidance associated with social anxiety. The average LSAS across participants of both studies was 29.02, their average earning 1.93 Euros, and the slope $-0.008$ Euros/LSAS point; therefore a person with an LSAS of 0 would earn 1.93 + 0.008*29 = 2.162 Euros or 12% more than the average participant, while a person with an LSAS of 60, likely to suffer from generalised social phobia (*Rytwinski et al., 2009*), would earn 1.93–0.008*31 = 1.682 Euros or 12.8% less than the average participant and 22.2% less than a person with an LSAS of 0.

## Neuroimaging results

### Decision can be decoded from amygdala activation

To examine the involvement of the amygdala in the decision-making process, we tested whether neural activity in the amygdala during the decision phase could predict trial-by-trial decisions. To this end, we trained a linear support vector machine classifier with default parameters on patterns of BOLD response in both amygdalae to discriminate between safe and risky choices against both kinds of partners (searchlight MVPA with leave-one-run-out cross-validation; see Materials and methods). The data from 39 participants were included in this analysis, one was excluded because of

insufficient BOLD signal in the left amygdala. Decisions could indeed be decoded slightly above chance from activation in the basolateral subregion of the right amygdala (mean accuracy = 56.4%; MNI coordinates of peak: x = 28, y = −4, z = −16, Z = 4.05, 51 voxels, FWE-corrected at p<0.05 for multiple comparisons at the cluster level within anatomically-defined amygdalae, based on a voxel-wise threshold of p<0.001 uncorrected; *Figure 3A*). Decoding accuracy did not vary with social anxiety ($F(1,36)$ = 0.1, p=0.76, $R^2$=0.00). Although decoding accuracy was relatively low, these findings are compatible with the involvement of a part of the amygdala in the decision-making process engaged by our task.

## Activation during decision and outcome

We focused on investigating the contributions of the amygdala and ventral striatum to the neural substrates underlying the tendencies to seek or avoid social situations. We used the BOLD responses obtained during the outcome phase of the trial to identify regions of interest (ROIs) sensitive to the human partner and to wins vs. no wins. Activation in bilateral amygdalae was higher in response to feedback from the human partner compared to feedback from the computer partner (Left: MNI coordinates x = −20, y = −6, z = −16, Z = inf., 143 voxels, p<0.001; right: MNI x = 20, y = −8, z = −16, Z = inf., 163 voxels, p<0.001; *P* values FWE-corrected for multiple comparisons at the voxel level across the whole brain; *Figure 3B*). 9% of the right amygdala cluster lay within the cluster identified in the decoding analysis reported above, 29% of the decoding analysis cluster lay within the right amygdala cluster described here, and 7.5% of the voxels of these clusters were common to both. Activation in bilateral ventral striatum was higher in response to wins vs. no wins (Left: MNI coordinates x = −10, y = 10, z = −8, Z = 3.84, p=0.014; right: MNI x = 12, y = 10, z = −8, Z = 4.03, p=0.009; *P* values FWE-corrected for multiple comparisons within ventral striatum; *Figure 3C*). We extracted parameter estimates from these ROIs to assess whether BOLD responses varied as a function of trait social anxiety (=LSAS score) or value of social engagement ($CE_{50\ Human} - CE_{50\ Computer}$) at both the decision and outcome stages of the task. These tests were independent of the contrast used to define the regions of interest, and data from each individual participant were extracted from ROIs defined in a group analysis run on the data of all the other participants (*Esterman et al., 2010*).

## Activation during the decision: Amygdala

Participants' amygdala BOLD activation was higher when deciding whether to engage in the game against a human rather than a computer (main effect of human vs. computer opponent: left

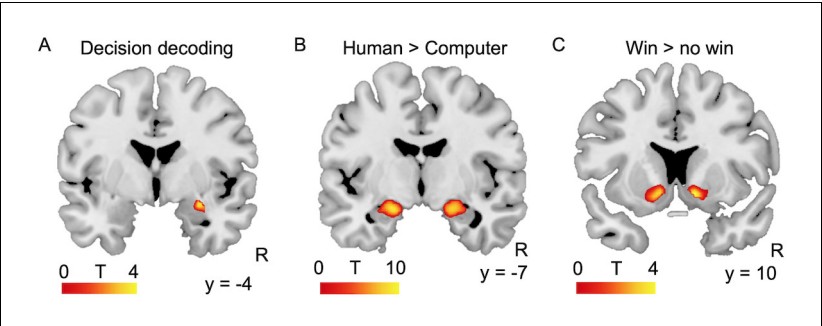

**Figure 3.** Amygdala and ventral striatum activation during the task. (**A**) Decision (safe or risky) could be decoded with 56.4% accuracy from amygdala activation. Decoding accuracy map resulting from an MVPA searchlight analysis of the BOLD signal obtained at the time of the decision, thresholded at p<0.05 FWE-corrected across anatomically-defined amygdalae. (**B**) Amygdala response to game outcome: response to feedback from the human partner (facial expression) minus the response to feedback from the human partner (abstract symbol). Threshold was p<0.05 FWE-corrected for multiple comparisons at the voxel level across the whole brain. (**C**) Clusters in bilateral nucleus accumbens showed significant activation increases in response to wins compared to no wins. Threshold was p<0.05 FWE-corrected for multiple comparisons at the voxel level within the anatomically-defined ventral striatum. R: right; y = MNI coordinate.
DOI: https://doi.org/10.7554/eLife.45249.005

amygdala: $F(1,38) = 6.24$, p=0.017, $\eta_p^2$=0.141, n = 39 participants as one had insufficient BOLD signal in the left amygdala ROI; right amygdala: $F(1,39) = 5.07$, p=0.030, $\eta_p^2$=0.12). While there was no significant difference between risky and safe decisions (left amygdala: $F(1,38) = 1.85$, p=0.18, $\eta_p^2$=0.05; right amygdala: $F(1,39) = 2.71$, p=0.11, $\eta_p^2$=0.07), importantly there was a significant interaction between partner and decision (left amygdala: $F(1,38) = 11.61$, p=0.002, $\eta p^2$=0.234; right amygdala: $F(1,39) = 13.83$, p<0.001, $\eta_p^2$=0.262). Post-hoc tests revealed a stronger response to humans than computers when participants chose the risky option (left amygdala: $t(38) = 3.35$, p=0.002, $d = 0.54$; right amygdala: $t(39) = 3.57$, p=0.001, $d = 0.56$), and no significant difference between partners when participants chose the safe option (left amygdala: $t(38) = 1.11$, p=0.28, $d = 0.17$; right amygdala: $t(39) = 0.65$, p=0.52, $d = 0.10$, see *Figure 4A and B*).

To relate the amygdala response to our behavioural measures, we assessed whether the interaction just described [(Risky $_{Human}$ - Risky $_{Computer}$) - (Safe $_{Human}$ - Safe $_{Computer}$)] varied with LSAS or our experimental estimate of the value of social engagement ($CE_{50\ Human} - CE_{50\ Computer}$). The left amygdala response showed influences of both variables (LSAS: $F(1,37) = 4.67$, p=0.037, $R^2$=0.12; social engagement: $F(1,37) = 5.49$, p=0.025, $R^2$=0.13, partial regression plot see *Figure 4C*), but the right amygdala response did not ($F(1,38) < 2.6$, p>0.12, $R^2$<0.07).

## Activation during the outcome: Amygdala

At the time of gamble outcome, the left amygdala response difference to outcomes of games (winning and losing) against human compared to computer partners increased with social anxiety ($F(1,37) = 5.89$, p=0.02, $R^2$=0.14; *Figure 4D*) and decreased with the value of social engagement ($F(1,37) = 6.03$, p=0.02, $R^2$=0.14; *Figure 4E*). These effects did not quite reach significance in the right amygdala data (LSAS: $F(1,38) = 1.74$, p=0.20, $R^2$=0.05; social engagement: $F(1,38) = 3.88$, p=0.056, $R^2$=0.10). The main effect of partner was not tested as the response difference between human and computer was the contrast used to define these ROIs. There was no significant difference between amygdala responses to wins or losses, nor interaction between outcome and partner, nor variations in these effects as a function of anxiety or value of social engagement (all $F < 3.2$, p>0.08).

## Activation during the outcome: Nucleus accumbens

Right nucleus accumbens response to wins varied as a function of the partner and participants' anxiety level: the response was higher when participants won against the human rather than the computer ($F(1,36) = 5.26$, p=0.028, $\eta_p^2$=0.13; n = 38 participants, two were excluded because of insufficient BOLD signal) and this difference decreased with increasing social anxiety traits ($F(1,36) = 5.59$, p=0.024, $R^2$=0.13, *Figure 4F*). This effect was not significant in the left NAcc response ($F(1,37) = 2.29$, p=0.14; $n = 39$ participants, one was excluded because of insufficient BOLD signal). Responses to 'no win' outcomes, and differences between not winning against a human vs. a computer did not vary with social anxiety; none of these variables varied with the value of social engagement (all $F < 1.2$, p>0.2, $R^2$<0.04). The main effect of outcome (win vs. no win) was not tested as the response difference between wins and no wins was the contrast used to define these ROIs.

## Functional connectivity during the decision

To investigate whether social anxiety traits influenced the functional connectivity between amygdala and nucleus accumbens during social decision-making, we performed a psychophysiological interaction analysis on the data obtained at the time of the decision (*Friston et al., 1997*; *McLaren et al., 2012*). Using our ROIs as seeds, we first examined functional connectivity between these ROIs. We extracted parameter estimates for the psychophysiological regressors based on the nucleus accumbens signal at the time of the decision from the amygdala ROIs (data were obtained from n = 38 participants). We investigated connectivity in the critical interaction [difference between human vs computer in the risky decision, minus the difference between human vs computer in the safe decision: (Risky $_{Human}$ - Risky $_{Computer}$) - (Safe $_{Human}$ - Safe $_{Computer}$)]. We found that connectivity between right nucleus accumbens and both the left and right amygdalae increased with social anxiety (right NAcc - left amygdala: $F(1,36) = 5.5$, p=0.025, $R^2$=0.13, *Figure 5A*; right NAcc - right amygdala: $F(1,36) = 12.56$, p=0.001, $R^2$=0.26, *Figure 5B*). These effects were not found in the connectivity between left nucleus accumbens and the amygdalae (all $F(1,36) < 1.7$, all p>0.2, all $R^2$<0.06). None

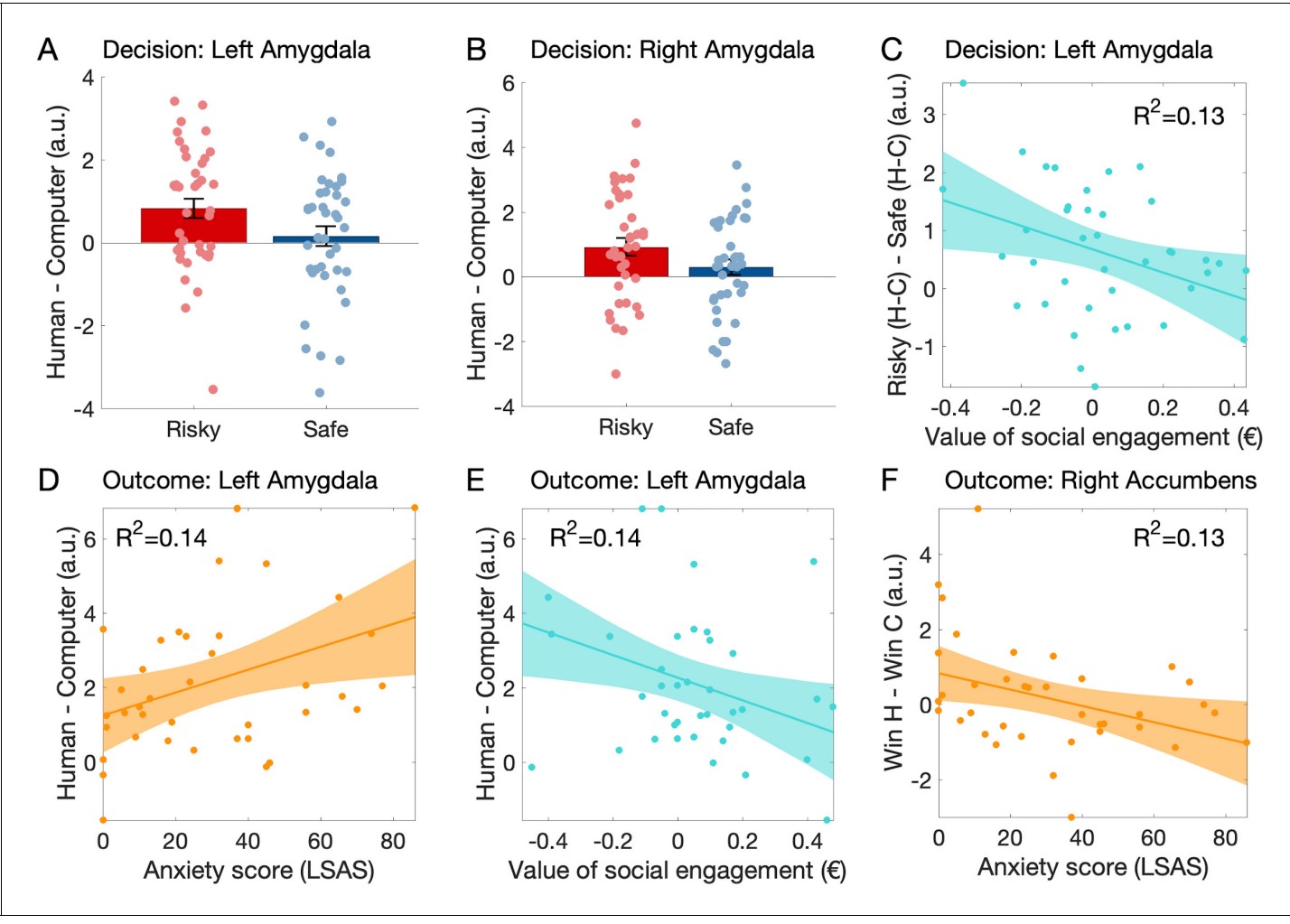

**Figure 4.** Variation of the neural activation as a function of choice, partner, social anxiety level (LSAS score) and the value of social engagement ($CE_{50\ human} - CE_{50\ computer}$) during the decision. (A–C) and the outcome (D–F) phases of the experiment. (A) During the decision phase of the trial (no face was shown), the left amygdala response was higher when participants chose the risky option against the human rather than the computer; this difference was not found when participants chose the safe option. (B) Similar effects were observed in the right amygdala. (C) In the left amygdala, the response difference between risky choices with human vs. computer partners and safe choices with human vs. computer partners (interaction: risky - safe choices X human - computer) decreased with the measured value of social engagement. (D) During the outcome stage of the game, the left amygdala response to outcomes of gambles against a human (=emotional expressions) compared to a computer increased with LSAS and (E) decreased with the measured value of social engagement. (F) In the right nucleus accumbens, the response difference between winning against a human partner rather than a computer partner decreased with LSAS. A.u.: arbitrary units. In panels A and B, central tendency is the mean and indicated by bar height, error bars indicate standard error of the mean. Grey-shaded areas represent the 95% confidence intervals around the slope of the regression line.

DOI: https://doi.org/10.7554/eLife.45249.006

of these connectivity measures varied with the value of social engagement (all $F < 3.3$, all $p>0.07$, all $R^2<0.08$).

We then examined functional connectivity between our ROIs and the rest of the brain. We searched for regions showing connectivity increases when participants decided to engage in the social situation (interaction contrast described above), and identified three clusters with significant effects ($p<0.05$ FWE-corrected across the whole brain at the voxel level): dorsal anterior cingulate connected to left amygdala (Peak: MNI coordinates $x = -2$, $y = 16$, $z = 40$, Brodmann Area (BA) 24a/b, $Z = 5.28$, $p=0.01$); cuneus/medial occipital cortex connected to left nucleus accumbens ($x = -2$, $y = -62$, $z = 8$, BA 17, $Z = 5.32$, $p=0.007$), and the perigenual part of the anterior cingulate cortex connected to right nucleus accumbens (henceforth called pACC; $x = 0$, $y = 36$, $z = 14$; BA 24a/b,

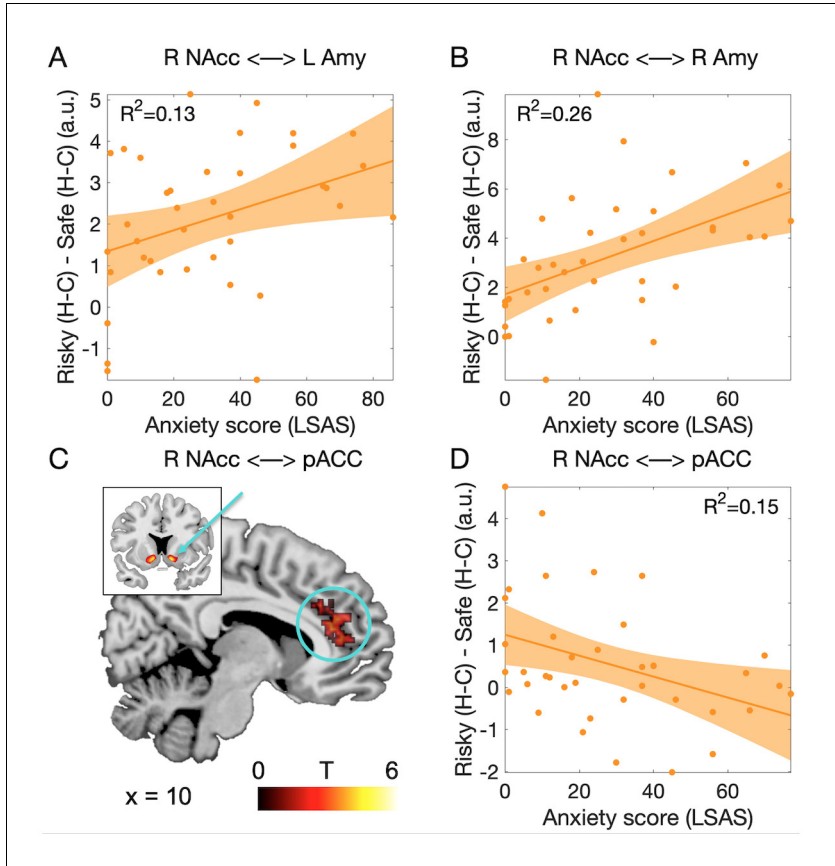

**Figure 5.** Functional connectivity associated with the decision to engage in the social situation. (**A**) Connectivity between left NAcc and right amygdala increased with the level of social anxiety. (**B**) Connectivity between right NAcc and right amygdala showed the same effect. (**C**) Cluster of voxels in the perigenual anterior cingulate cortex (pACC, BA24a/b; circle) showing functional connectivity modulation with right nucleus accumbens (arrow) as a function of the partner and social anxiety. Threshold was p<0.05 FWE-corrected for multiple comparisons across the whole brain at the cluster level (based on p<0.001 uncorrected threshold, minimum cluster size = 100 voxels). (**D**) Connectivity between right NAcc and pACC decreased proportionally to the level of social anxiety. Abbreviations as in *Figures 3* and *4*; pACC: perigenual anterior cingulate.

DOI: https://doi.org/10.7554/eLife.45249.007

$Z$ = 5.36, p=0.005, *Figure 5C*). We then examined whether connectivity in any of these clusters varied with social anxiety traits, and found significant effects in the pACC only ($F(1,36)$ = 6.42, p=0.016, $R^2$=0.15, *Figure 5D*).

## Discussion

This study shows that trait social anxiety is associated with reduced subjective valuation of engaging in a social situation, and amygdala and ventral striatum activation and functional connectivity differences related to social anxiety during social decision-making, both at the decision stage and when experiencing the outcome of a social situation. Interestingly, both relatively more anxious and relatively more sociable participants deviated from economically optimal decisions in order to either seek (more sociable participants) or avoid (more anxious participants) the social situation. This mirrors real-life behaviour, where sociable people spend money to interact with other people whereas socially anxious participants may instead spend money in order to avoid social interactions. Crucially, the observed behavioural effects were (i) specific to interactions with a human partner and thus not due to general risk aversion, which is increased in anxiety (*Hartley and Phelps, 2012*), (ii) not related to differences in the subjective perception of the human feedback stimuli as a function of anxiety,

and (iii) specific for social anxiety (measured with the LSAS) and not with measures of related but non-specific traits such as general anxiety (STAI-T), depression (BDI-II) or autistic traits (AQ).

BOLD signal data acquired during this task revealed that amygdala activation may be related to the decision-making process: participants' safe or risky choices could be decoded from activation at the time of decision-making in the right amygdala. These findings are compatible with the previously reported involvement of the primate amygdala in social decision-making (*Chang et al., 2015*; *Grabenhorst et al., 2013*). While decoding accuracy did not vary with social anxiety, amygdala activation, both at the time of social decision-making and during the outcome phase of the social situation (i.e. when participants experienced social feedback), varied with social anxiety and with the value of social engagement estimated prior to the scan. These findings are compatible with findings of increased amygdala responses to threat stimuli in people with higher trait anxiety (e.g. *Etkin et al., 2004*) and extend this association to the domain of social decision-making.

Nucleus accumbens response to receiving positive social feedback, that is winning the game of dice against the human compared to winning against a computer partner, decreased with social anxiety. In contrast, no effects of social anxiety were found on the response to 'no win' outcomes. These results are compatible with previous findings indicating that social anxiety is associated with reduced striatal response to different kinds of social rewards (*Richey et al., 2014*; *Sripada et al., 2013*).

Social anxiety influenced the functional connectivity between the left nucleus accumbens and both amygdalae, and between right nucleus accumbens and the perigenual anterior cingulate cortex (pACC). Connectivity between amygdala and nucleus accumbens during the decision to engage in the social interaction increased with social anxiety. These findings are consistent with the importance of this connection for avoidance behaviour: Disruption of the (basolateral) amygdala - nucleus accumbens (shell) projection in rats has been shown to impair avoidance behaviour (*O'Doherty et al., 2004*). Our findings suggest that in humans, functional connectivity between these regions when engaging in a social interaction is greater in people with more pronounced social anxiety. In contrast, nucleus accumbens – pACC connectivity decreased proportionally with increasing social anxiety. Several previous studies have reported a reduced functional connectivity in social anxiety between parietal, limbic and executive network regions during resting state or in response to presentation of emotional faces (*Brühl et al., 2014*). The pACC is functionally dissociable from subgenual anterior cingulate (*Pezawas et al., 2005*; *Apps et al., 2016*), connected to ventral striatum (*Kunishio and Haber, 1994*; *Margulies et al., 2007*) and reduced in volume in people with a genotype associated with increased anxiety-related temperamental traits and risks for depression (*Pezawas et al., 2005*). pACC has also been associated with the response to social stress and negative social feedback (*Dedovic et al., 2009*; *Lederbogen et al., 2011*) as well as with specific social cognitive functions such as the tracking of other people's motivation (*Apps et al., 2016*). A dysfunction of pACC and its regulatory influence on amygdala and ventral striatum has even been associated with poor response to chronic social defeat, a possible pathophysiological mechanism of schizophrenia (*Selten et al., 2017*). Our findings are compatible with the idea that these differences in regulatory influence of pACC on subcortical circuits are also observable during social decision-making.

Our neuroimaging findings described above reveal a multifaceted variation in the subcortical neural correlates of both decision-making and processing of social feedback associated with social anxiety and social avoidance. These response variations are compatible with increases of threat-related responses, reduced social reward signals, increased avoidance-related network activity and reduced top-down feedback on these networks. While causal relationships between these findings cannot be assessed, all may play a role in social avoidance and social anxiety. Interestingly, the combination of higher threat-related amygdala and lower reward-related ventral striatum activity have recently been associated with increased risk of developing alcohol use disorder in response to stress (*Nikolova et al., 2016*). As engaging in social decisions can be stressful for many people, paradigms such as ours may allow to investigate the link between social decisions, social stress and mental disorders. Further studies specifically designed to investigate connectivity between these structures will be required to elucidate the details of the neural mechanism underlying social avoidance.

Avoidance behaviour was not reflected in explicit valence judgments of the feedback stimuli. Our effects thus seem not to have been driven by differences in the conscious interpretation of the feedback but rather by differences in the desire to expose oneself to such feedback. While reduced explicit approachability ratings for positive facial expressions have been reported in social anxiety

(*Campbell et al., 2009*), socially anxious individuals in another study have shown avoidance behaviour without reduced valence ratings (*Heuer et al., 2007*). It thus appears that while implicit measures often show avoidance, effects on explicit measures may be task-dependent (*Staugaard, 2010*).

If our task can be taken as an example of behaviour in real-life social interactions, it may be used to quantify one aspect of the costs resulting from the avoidance behaviour observed in social anxiety. Our linear regression results suggest that persons likely to suffer from generalised social phobia would earn about 22% less in this game than people with an LSAS score of 0. It may be interesting to combine our experimental approach with estimations based on patient's costs related to medical care or productivity loss (*Wittchen et al., 2000*; *Patel et al., 2002*; *Stein et al., 2005*; *Acarturk et al., 2009*) in future studies. It may also be interesting to run our study on patients with diagnosed social anxiety to evaluate whether our task could be useful as an experimental rather than a self-report-based measure to identify socially anxious individuals, an advance aligned with the NIMH's RDoC proposal (*Insel, 2014*).

There are several limitations to the work presented here. First, the number of participants in both studies is relatively low for the assessment of interindividual differences. While the sample size of Study two roughly corresponds to the number of participants recommended by a power analysis based on the results of Study 1 (see Materials and methods), effect size estimates for inter-individual differences based on such a relatively low number of participants are bound to be imprecise (see size of the confidence bounds in *Figures 2*, *4* and *5*) (*Schönbrodt and Perugini, 2013*). One should thus be careful when interpreting our results, including our estimates of the costs of social avoidance. Another issue concerns our study sample: our participants were young German participants, most of them students at the local university, who were willing to subject themselves to behavioural and neuroimaging experiments performed by unknown experimenters; such situations are likely to be quite distressing for individuals with severe levels of social anxiety. This restriction is reflected in the limited range of LSAS scores of our sample: only 11% of participants showed LSAS levels found in people suffering from generalised social phobia (>60). We must thus exercise caution when drawing inferences about clinical populations based on our results. Next, while we could significantly decode participants' choice from activation in the right amygdala, the average accuracy was quite low. Therefore, caution must be used in interpreting this finding: the amygdala cluster identified in our analysis is unlikely to be the major contributor to participants' choices. Further studies specifically designed for a decoding analysis and investigating additional brain regions are required to better understand the neural mechanism underlying the decision-making process in our task. Despite these considerations, the fact that we could replicate the link between social anxiety and our experimental measure of the value of social engagement across two studies, with almost identical slope estimates, suggests that our approach has identified an interesting link between a relevant but subjectively reported, real-world behavioural trait and a controlled experiment.

## Materials and methods

### Ethics

The studies fulfilled all relevant ethical regulations and were approved by the local ethics committee of the Medical Faculty of the University of Bonn, Germany. All subjects gave written informed consent and the studies were conducted in accordance with the latest revision of the Declaration of Helsinki. Subjects were remunerated for their time (10 Euros/hour) and received game earnings (0–6 Euros).

### Participants

68 healthy participants (24 male, mean age 25.2, range 20 to 37) participated in Study 1 (behaviour only) and 47 healthy participants (19 male, mean age 28.5, range 22 to 40) participated in Study 2 (behaviour and fMRI). Participants were recruited from the local population through advertisements on online blackboards at the University of Bonn and on local community websites, and through flyers posted in libraries, university cafeterias and sports facilities (recruitment period: May 2016 – August 2017). The number of participants recruited in Study two roughly corresponds to the sample size estimated for a point biserial model test based on the results of Study 1 [$R^2$ = 0.13, one-tailed test with alpha error = 0.05 and power (1-beta)=0.8, sample size = 43; G*Power 3.1 (*Faul et al., 2007*)].

Participants were remunerated for their time (10 Euros/hour) and received game earnings (0–6 Euros). The data from seven participants in Study two were excluded from the fMRI data analysis: five participants moved their head too much for reliable motion correction (>3 mm or >3°) and two were excluded because they never choose the risky option in trials with human partners, prohibiting the analysis of neural responses during this condition. Data of individual participants were excluded from the ROI analysis if less than 10 voxels were included in their first-level GLM mask for the ROI considered, indicating insufficient MR signal quality. All subjects gave informed consent and the ethics committee of the Medical Faculty of the University of Bonn, Germany approved all studies.

## Studies

Study 1 consisted of one behavioural experiment, Study 2 (different participants) consisted of the same behavioural experiment and a subsequent fMRI experiment.

## Experimental task

The task was inspired by previous studies quantifying the value of social stimuli (*Deaner et al., 2005*). It was implemented in MATLAB (Version R2016b; RRID:SCR_001622; The MathWorks, Inc, Natick, MA) using the Psychtoolbox extensions (RRID:SCR_002881; http://psychtoolbox.org). Participants decided on each trial whether to play a game involving a gamble against a partner (risky option) or not (safe option) (*Figure 1A and B*). The game was played in two separate blocks of 126 trials against human partners (*Figure 1C*; 21 trials with each of the six partners) and a computer partner. Each trial proceeded as follows (*Figure 1A*): The risky option (gamble against the partner) was shown as an image on one side of the screen, and the safe option (amount of money obtainable as an alternative) was shown on the other side of the screen (screen sides were chosen randomly on each trial). Participants then chose the risky or safe option under no time pressure. If the participant chose the risky option, virtual dice were rolled for the participant and the partner, and the higher number won (50% winning chance, explicitly stated to participants; equal numbers led to immediate repetition of the dice roll, not displayed). The outcome was revealed after a delay of 1 s: if the participant won, they received 3 Euros, if they did not, they received 0 Euros. Importantly, the participant witnessed the partner's reaction (=feedback) to the outcome of the game at the same time. Feedback from human partners consisted in videos (duration 1 s) of facial expressions of admiration in case of winning, and condescension in case of no win. As in previous neuroimaging studies (*Schultz and Pilz, 2009*; *Schultz et al., 2013*), we used videos of facial expressions instead of pictures as feedback because such stimuli have higher ecological validity and lead to improved recognition of emotions (*Krumhuber et al., 2013*): dynamic displays of emotions are judged as more natural (*Sato and Yoshikawa, 2004*), lead to higher judgments of intensity and arousal (*Biele and Grabowska, 2006*), improve the recognition of subtle facial expressions (*Ambadar et al., 2005*; *Cunningham and Wallraven, 2009*) and evoke stronger and more widespread neural responses than static faces (*Schultz and Pilz, 2009*; *Schultz et al., 2013*; *Puce et al., 1998*; *Sato et al., 2004*; *Fox et al., 2009*; *Arsalidou et al., 2011*), particularly in brain areas associated with social cognition (*Schultz et al., 2013*; *Puce et al., 1998*; *Furl et al., 2007*). The facial expressions were selected from a validated database (*Kaulard et al., 2012*). Feedback from the computer partner were abstract symbols (green 'check' mark for win or red 'X' for no win). Perceived valence ratings about all feedback stimuli were collected after the experiment using a visual analogue scale (range 0–100). The safe option consisted in an amount of money varied between 0 and 3 Euros across trials (0, 0.5, 1, 1.5, 2, 2.5, or 3 Euros, equal probability, random order, 18 trials per amount). The game's certainty equivalent (termed $CE_{50}$; equivalent to the game's subjective value) was defined as the amount of payoff a participant would have to receive to be indifferent between that payoff and the game, and was estimated by fitting a cumulative Gaussian function to each participant's choice probabilities observed for the amounts offered in the safe option (*Figure 2A and B*). This $CE_{50}$ (and in Study 2, the amounts needed to obtain 20% and 80% safe choices, $CE_{20}$ and $CE_{80}$) was calculated for the computer and for the human partners. One trial was randomly selected and paid out at the end of the experiment. Under the simplest assumption (i.e. neutral attitude towards risk, choose 'play' if alternative amount is less than 1.5 Euros, 'play' or 'no play' at equal rates if alternative amount is 1.5 Euros, and 'no play' if alternative amount is above 1.5 Euros), the average earning per trial (and thus per game, as only one trial was ultimately rewarded) would be 1.93 Euro.

In Study 2, participants first performed the same experiment as described above and were then scanned while performing a modified version of the task immediately after the behavioural experiment. The fMRI task included the following modifications: (i) partner (computer or human) was determined randomly on each trial; (ii) at the decision stage, participants were presented with names (first names associated with the humans, or the word 'computer') instead of images; (iii) the alternative monetary amounts varied between the three possible values $CE_{50}$, $CE_{20}$ and $CE_{80}$, determined individually, in order to equate the number of safe and risky choices across individuals with different anxiety levels; (iv) the temporal intervals between trials and between phases of the trail (decision stage and outcome) were varied between 2 and 11 s following a Gamma distribution to dissociate the responses in the different parts of the trial. There were 72 trials per scanning run (36 per human and computer partner), and two runs per participant.

## Personality traits

Social anxiety trait levels of each participant using the self-report version of the Liebowitz Social Anxiety Scale [LSAS, (*Rytwinski et al., 2009*)]. Additional traits were obtained using Beck's Depression Index [BDI, (*Beck et al., 1996*)], Spielberger's Trait Anxiety Scale [STAI-T, (*Spielberger and Gorsuch, 1970*)] and the Autism-Spectrum Quotient [AQ, (*Baron-Cohen et al., 2001*)].

## Statistics

Statistics on the behavioural data and neural activation data from regions of interest (see below) were performed using JASP software (JASP Version 0.9.2; RRID:SCR_015823; JASP Team 2018; jasp-stats.org). As data were normally distributed, tests performed included multiple linear regression, repeated-measures ANOVA, and t-tests, including normality tests (Q-Q plot; Kolmogorov-Smirnov goodness-of-fit test). Whole-brain activation statistics were performed with SPM12 software (RRID:SCR_007037; Wellcome Trust Centre for Neuroimaging, London, UK; http://www.fil.ion.ucl.ac.uk/spm) running in MATLAB with corrections for multiple comparisons (for details, see 'fMRI data analysis' below). All statistical tests were two-tailed. Bayes factors were calculated using default priors and express the probability of the data given H1 relative to H0 (BF10, values larger than one are in favour of H1). Effect sizes were calculated using standard approaches implemented in JASP software.

## fMRI data acquisition and preprocessing

Imaging data were collected on a 3T Siemens TRIO MRI system (Siemens AG, Erlangen, Germany) with a Siemens 32-channel head coil. Functional data were acquired using a T2* echo-planar imaging (EPI) BOLD sequence, with a repetition time (TR) of 2500 ms, an echo time (TE) of 30 ms, 37 slices with voxel sizes of $2 \times 2 \times 3$ mm$^3$, a flip angle of 90°, a field of view of 192 mm and PAT two acceleration. To exclude subjects with apparent brain pathologies and facilitate normalisation of the functional data, a high-resolution T1-weighted structural image was acquired, with a TR of 1660 ms, a TE of 2540 ms, 208 slices with voxel sizes of $0.8 \times 0.8 \times 0.8$ mm$^3$ and a field of view of 256 mm. Data were then preprocessed and analysed using standard procedures in SPM12. The first five volumes of each functional time series were discarded to allow for T1 signal equilibration. The structural image of each participant was coregistered with the mean functional image of that participant. Functional images were corrected for head movement between scans by a 6-parameter affine realignment to the first image of the time-series and then re-realigned to the mean of all images. The structural scan of each participant was spatially normalised to the current Montreal Neurological Institute template (MNI305) by segmentation and non-linear warping to reference tissue probability maps in MNI space, and the resulting normalisation parameters were applied to all functional images which were then resampled at $2 \times 2 \times 2$ mm$^3$ voxel size, then smoothed using an 8 mm full width at half maximum Gaussian kernel. Time series were de-trended by the application of a high-pass filter (cut-off period, 128 s). For the multivariate decoding analysis, data preprocessing was identical except for omission of the normalisation and smoothing steps.

## fMRI data analysis

Functional data were analysed using a two-stage approach based on the general linear model (GLM) implemented in SPM12: individual participants' data were modelled with a fixed effects model, and

their summary data were entered in a random effects model for group statistics and inferences at the population level. For the main (i.e. non-decoding) analysis, the fixed effects model implemented a mass univariate analysis applied to normalised data. For the decoding analysis, the same fixed effects model was applied to non-normalised data, and patterns of parameter estimates were used for a multivariate (multivoxel) analysis. The accuracy maps resulting from the decoding analysis were then normalised to MNI space and entered into a random effects model for group statistics.

The fixed-effects model included the following event types per session: decision to play or not and win or loss outcomes, all modelled separately for the human or computer partner, resulting in eight event types. Regression coefficients (parameter estimates) were estimated for each voxel of each participant's brain. For the main (univariate) analysis, linear contrasts were applied to the individual parameter estimates of the response to the experimental conditions, resulting in contrast images. These were subjected to a group-wise random effects ANOVA in order to identify brain regions sensitive to the human partner, and brain regions sensitive to reward, using the BOLD responses collected during the outcome phase of the trial. The former regions were identified by subtracting responses to trials with computer partners from responses to trials with human partners, using a significance threshold of p<0.05, with family-wise error correction for multiple comparisons (FWE) at the voxel level across the whole brain. The latter, reward-sensitive regions were identified by subtracting responses to losses from responses to wins in the region of interest, using a threshold of p<0.05 FWE-corrected across voxels in the anatomically-defined ventral striatum, as provided in the WFU Pickatlas toolbox for SPM (RRID:SCR_007378; http://fmri.wfubmc.edu/software/pickatlas) (*Maldjian et al., 2003*). We restricted our analyses to these regions and to the regions functionally connected to them.

Parameter estimates of the response during the decision phase of the trials were then extracted from these regions of interest using MATLAB, as follows. All parameter estimates used for further analysis were conducted based on a leave-one-out procedure (*Kriegeskorte et al., 2009*) to avoid non-independence bias in data analysis. Specifically, we ran n = 38 leave-one-subject-out group-level GLM analyses, and each of these GLMs was used to define the clusters of interest for the subject left out (*Esterman et al., 2010*). To further reduce circularity or 'double-dipping' issues, the contrasts used to define the clusters (i.e., outcome human >outcome computer for the human-sensitive regions and win >loss for the reward-sensitive regions) were not calculated on the extracted data. The influence of anxiety traits on these neural responses was then assessed using linear regression models implemented in JASP, with social anxiety level as independent variable and parameter estimates as dependent variable.

## Decoding analysis

We used the Decoding Toolbox (https://sites.google.com/site/tdtdecodingtoolbox/, version 3.991) for SPM, applied directly to the single-subject fixed effects model described above (data were not normalised and not smoothed). Independent variables were the decision (play or not) and the partner (human or computer). Features were the parameter estimates for the corresponding regressors in the model; no dimensionality reduction was used. We ran a classification searchlight analysis on each participant's data, designed to decode the decision to play or not (chance accuracy = 50%), using a support vector machine classifier with default parameters (LIBSVM; RRID:SCR_010243; parameters: C = 1, type C-SVC, linear kernel), 9 mm search radius, trained on the data of one run and tested on the data of the other run. The resulting individual decoding accuracy maps minus chance were normalised to MNI space and entered into a second-level random effects t-test analysis against 0 in SPM, designed to select voxels with reliable above-chance accuracy values in the participant group. Results were restricted to the ROIs (amygdalae and bilateral nucleus accumbens, defined anatomically using the WFU Pickatlas toolbox), and thresholded at p<0.05, with family-wise error correction for multiple comparisons at the voxel level within the ROIs. For tests of the effects of social anxiety on decoding accuracy, accuracy scores were averaged within each ROI and compared across participants using linear regression.

## Psychophysiological interaction (PPI) analysis

We performed a PPI analysis following the standard procedure in SPM8, using the gPPI toolbox (*McLaren et al., 2012*) (version 13.1, http://www.nitrc.org/projects/gppi). We extracted BOLD

signals (eigenvariates) from our regions of interest (amygdala and nucleus accumbens clusters identified as responding more to the human than to the computer partner; see above). We then constructed, for each region of interest, another GLM for the PPI analysis identical to the GLM described above, including 1) the same psychological regressors ($n = 8$ per session); 2) the BOLD signal from the region of interest as a physiological factor ($n = 1$ per session); 3) a set of psychophysiological interaction (PPI) factors, which are an interaction of the deconvolved BOLD signal in the region of interest and the psychological factors of interest (we limited the analysis to the decision phase of the trial; $n = 4$ conditions per session); 4) and movement parameters estimated during motion correction included as confound regressors. All of the regressors except for the physiological factors and the movement parameters were convolved with a canonical HRF. For each participant, regression coefficients of the PPI factors were estimated for each voxel of each participant's brain, as in the primary GLM analysis described above. Linear contrasts were applied to the individual parameter estimates of the response to the PPI regressors, resulting in contrast images. These were subjected to a group-wise random effects ANOVA in order to identify brain regions showing a change in connectivity with the region of interest at the time of decision-making as a function of the partner, decision taken and participant anxiety level. The threshold used to identify clusters of interest was $p<0.05$, with family-wise error correction for multiple comparisons at the cluster level across all the voxels of the brain based on an uncorrected threshold of $p<0.001$ at the voxel level. Parameter estimates of the PPI regressors based on the nucleus accumbens activation were extracted from the amygdala ROIs in order to assess the connectivity between these structures using MATLAB.

The data that support the findings of this study are available from the corresponding author upon reasonable request.

## Acknowledgements

The authors would like to thank Franny B Spengler for help during the development of the experiment.

## Additional information

### Funding
The authors declare that there was no funding for this work.

### Author contributions
Johannes Schultz, Conceptualization, Resources, Data curation, Software, Formal analysis, Supervision, Validation, Investigation, Visualization, Methodology, Writing—original draft, Project administration, Writing—review and editing; Tom Willems, Maria Gädeke, Ghada Chakkour, Alexander Franke, Investigation, Project administration, Writing—review and editing; Bernd Weber, Conceptualization, Writing—review and editing; Rene Hurlemann, Conceptualization, Funding acquisition, Writing—review and editing

### Author ORCIDs
Johannes Schultz  https://orcid.org/0000-0003-4117-232X

### Ethics
Human subjects: All subjects gave written informed consent and the ethics committee of the Medical Faculty of the University of Bonn, Germany approved all studies (Approval number: 098/18).

### Decision letter and Author response
Decision letter https://doi.org/10.7554/eLife.45249.013
Author response https://doi.org/10.7554/eLife.45249.014

## Additional files

### Supplementary files

• Supplementary file 1. Supplementary behavioural results. Four tables displaying additional results: *Supplementary file 1A and 1B* show descriptive statistics for questionnaire data from Study one and Study two respectively, *Supplementary file 1C and 1D* show all steps of a backward stepwise regression analysis for the experiment data of Study one and Study 2, respectively.
DOI: https://doi.org/10.7554/eLife.45249.008

• Transparent reporting form
DOI: https://doi.org/10.7554/eLife.45249.009

### Data availability

Data are freely available on Dryad, http://doi.org/10.5061/dryad.jq44b1r.

The following dataset was generated:

| Author(s) | Year | Dataset title | Dataset URL | Database and Identifier |
|---|---|---|---|---|
| Schultz J | 2019 | Data from: A human subcortical network underlying social avoidance revealed by an econometric task | https://doi.org/10.5061/dryad.jq44b1r | Dryad Digital Repository, 10.5061/dryad.jq44b1r |

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
