## [Decision Letter]

Thank you for submitting your article "A human subcortical network underlying social avoidance revealed by an econometric task" for consideration by *eLife*. Your article has been reviewed by three peer reviewers, and the evaluation has been overseen by a Reviewing Editor and Michael Frank as the Senior Editor. The reviewers have opted to remain anonymous.

The reviewers have discussed the reviews with one another and the Reviewing Editor has drafted this decision to help you prepare a revised submission.

Summary:

This is an interesting paper investigating the role of social anxiety in human decision making. The authors use simple decisions in the context of social feedback or no- feedback (i.e. computer). The behavioral paradigm includes an important and appropriate non-social control condition. The behavioral and neural effects are interesting and it appears that the data were analyzed carefully.

Essential revisions:

All reviewers indicated that this paper covers a very interesting and timely study. In particular they all liked the clever behavioral approach measuring of the "cost" of social anxiety. Nevertheless, they also indicated major issues that need careful attention.

In particular, the reviewers flagged that clarification is needed concerning subject exclusion policy, Furthermore, they indicated that the number of subjects is quite low for addressing individual differences. The authors should indicate how they identified the sample size (e.g. power analysis) and that this should be clearly mentioned as a limitation of the study. One suggestion was to focus more on behavior, and cast the imaging analyzes as more exploratory or 'hypothesis generating'. Finally, the reviewers identified problems with the median split analysis (see Rucker, McShane and Preacher, 2015). With respect to the decoding analysis, the reviewers were unclear, what it precisely adds to the study. Although the authors report that "decisions could indeed be decoded above chance from activation in the basolateral subregion of the right amygdala" the mean accuracy was 54.6% (chance = 50%), which provides very limited evidence. Finally, the authors should clarify how the behavioral results are connected to the imaging results or if not why there are important in their own right. Finally, the authors should describe in more detail how a correction for multiple comparison regarding the anxiety measures was implemented.

*Reviewer #1:*

This is a wonderful study examining the frequency of social versus monetary interaction and correlating this with social anxiety. The results are clear and the paper is well written. I do have one major concern that need to be addressed.

Excluding subjects based on in scanner patterns vs. out scanner patterns seems problematic. Are there any citations to support this method? Of course, if this is pre-registered, it is not a problem, but you need to convince people that this was not post-hoc. One way is to include the subjects and show the same results and then state that you wanted clean data.

*Reviewer #2:*

Despite its basic and translational importance, the mechanisms underlying behavioral avoidance in social anxiety remain poorly understood. Thus, the focus of this paper is timely, important, and likely to be of keen interest to a broad spectrum of basic and clinical researchers.

Key strengths include:

1) The 2 study approach, which enabled replication and extension of the core behavioral finding (higher LSAS, reduced economic value of social stimulation – taken as indicative of 'greater avoidance'.

2) Thoughtful experimental methodology and rigorous statistical techniques, including evidence that trait social anxiety is unrelated to other aspects of performance (e.g. general risk-taking/aversion).

3) "Translating" the regression model into intuitive values ("person with an anxiety score at the cut-off for generalized social phobia (60 points) would thus value engaging in the social situation 0.42 Euros less than a person with LSAS score of 0.").

4) Multi-method approach (indifference estimate, mass univariate imaging, gPPI, decoding).

Key limitations include:

1a) Only n=38 usable datasets (fMRI+LSAS+behavior) for the imaging study. Effect sizes for individual differences (social anxiety) analyses will be imprecise and unstable.

For example, the 95% CI's for a correlation of r=.35 in n=38 range from 0.035 (nil) to 0.602 (very strong).

See also recent work by Schönbrodt and Perugini (2013) titled "At what sample size do correlations stabilize?".

1b) This concern is amplified by the authors' use of a post-hoc median split collected from an unselected sample, which tends to further reduce power (decrease the precision of effect size estimates).

1c) This concern is mitigated for the behavioral effects by the between-study replication (cf. Figure 2C and D).

2) The value of the decoding analyses is unclear. Although the authors report that "decisions could indeed be decoded above chance from activation in the basolateral subregion of the right amygdala" the mean accuracy was 54.6% (chance = 50%), which provides very limited evidence. Moreover, it is unclear whether this cluster overlaps those moderated by social anxiety.

3) The imaging results and the behavioral results proceed in parallel and remain unconnected. Thus, claims that these results "suggest how activity differences in a subcortical network during social decision-making may lead to social avoidance" need to be tempered. Given the modest sample size, traditional statistical approaches for linking seem inadvisable.

– gPPI and other functional connectivity measures do not license the interpretation of "information transmission".

– The use of a post-hoc median split is contrary to published recommendations from Kris Preacher and other statisticians.

– Interactions needs to be fully decomposed and adequately explained.

– "In other words, given an average earning in the game of 1.93 Euros (see Materials and methods), that person would lose 21.8% of the average earning in the game."

"Our linear regression results suggest that persons likely to suffer from generalized social phobia (i.e. those with an LSAS score of 60) (21) would lose 21.8% of their earnings in this game."

This is misleading because the average earnings reflect the average LSAS, not 0.

Discussion/implications:

– Fails to adequately acknowledge key study limitations.

– The strength of the conclusions is not well calibrated to the strength of the evidence. The authors extrapolate a bit too optimistically based on a sample of 38 unselected young Germans, all of whom were willing to come to the lab and subject themselves to procedures that would be quite distressing to individuals with elevated levels of social anxiety (e.g. perform unfamiliar tasks in a novel setting while being scrutinized by unfamiliar experimenters).

*Reviewer #3:*

This paper studies a risk task in which subjects choose whether to take a same amount of money or gamble to win more or get less. In the human condition upon win/loss they will see an admiring or condescending (positive/negative) face. The amount they will give up in expected earnings to avoid these facial 'judgments' is linked to social anxiety.

There is a lot to like about this paper. There are two studies, the second replicating the first behaviorally (and is used for fMRI).

One challenge is that there are four social anxiety scales used (LSAS and AQ in study 1, and two others added in study 2). One has to correct for multiple comparison in using all these scales as failing to do so inflates the apparent significance of the "best" scale.

Admirable: study 1 and 2 replicate behavioral pattern. This is really great.

I like the use of an economic measure of the "cost" of social anxiety. As authors note at the tail end of the paper, one could do a lot of interesting pricing of disorders in this way. There probably are similar studies in economics, but often use inferior measures of disorder and of course it is hard to capture all the costs and benefits. In any case, it is great to see this calculation put forward here even though it is not ideal.

Not sure why the term "opponents" are used. Do these facial opponents gain/lose when the subjects lose/win?

The ppi works rather nicely.

---

## [Author Response]

Essential revisions:All reviewers indicated that this paper covers a very interesting and timely study. In particular they all liked the clever behavioral approach measuring of the "cost" of social anxiety. Nevertheless, they also indicated major issues that need careful attention.In particular, the reviewers flagged that clarification is needed concerning subject exclusion policy, Furthermore, they indicated that the number of subjects is quite low for addressing individual differences. The authors should indicate how they identified the sample size (e.g. power analysis) and that this should be clearly mentioned as a limitation of the study. One suggestion was to focus more on behavior, and cast the imaging analyzes as more exploratory or 'hypothesis generating'. Finally, the reviewers identified problems with the median split analysis (see Rucker, McShane and Preacher, 2015). With respect to the decoding analysis, the reviewers were unclear, what it precisely adds to the study. Although the authors report that "decisions could indeed be decoded above chance from activation in the basolateral subregion of the right amygdala" the mean accuracy was 54.6% (chance = 50%), which provides very limited evidence. Finally, the authors should clarify how the behavioral results are connected to the imaging results or if not why there are important in their own right. Finally, the authors should describe in more detail how a correction for multiple comparison regarding the anxiety measures was implemented.

We thank the Reviewing Editor for this summary of the main issues raised by the reviewers. Regarding subject exclusion policy: we have clarified and updated our policy and have re-analysed our fMRI data after including all the subjects that we could – see our reply to reviewer #1’s first comment below. We agree that we nevertheless still have relatively few subjects for a study on individual differences, the consequences of which we discuss in the new detailed limitations paragraph in the Discussions section; we also refer to a power analysis that supports our sample size for Study 2. Regarding the median-split analyses: we understand the issue raised by the reviewers and have completely removed median-split analyses from the paper by instead treating LSAS and the estimated value of social engagement as continuous variables. The accuracy of the decoding analysis is indeed low (although it has increased slightly to 56.4% after including subjects that were previously excluded); we now acknowledge the fact that it provides limited evidence, and do not base any major inferences on this result. We have now better connected the imaging and behavioral results by reporting a direct relation between the estimated value of social engagement and amygdala activation. Correction for multiple predictors in our regression models was addressed in two recommended ways. We have of course also addressed all further comments (see details below), including the addition of 4 supplementary tables, and have reworked our figures to hopefully present our results more clearly.

Reviewer #1:This is a wonderful study examining the frequency of social versus monetary interaction and correlating this with social anxiety. The results are clear and the paper is well written. I do have one major concern that need to be addressed.Excluding subjects based on in scanner patterns vs. out scanner patterns seems problematic. Are there any citations to support this method? Of course, if this is pre-registered, it is not a problem, but you need to convince people that this was not post-hoc. One way is to include the subjects and show the same results and then state that you wanted clean data.

We agree that we need to address this point much more carefully, especially given that this was not a pre-registered study. The exclusion was not based on a published method and was indeed post-hoc, based on changes in response patterns between the pre-scan behavioral session and the in-scan session. Following the present reviewers’ comments, including the very understandable concern regarding the low number of participants for a study of interindividual differences (see first comment by reviewer #2), we have decided to revise our exclusion policy, and now exclude only those participants whose data could not contribute to all experimental conditions. This was the case for two participants who did not choose the risky option with the human partner in either experimental run, thereby precluding analysis of the neural response during these decisions. This has allowed us to include 3 more subjects than in the original analysis. Our approach maximises the number of participants included in the analysis without resorting to imputation of missing data. Our results were only minimally affected by the data of these additional participants (see Results section). We now describe our exclusion policy in the Materials and methods section.

Reviewer #2:[…]Key limitations include:1a) Only n=38 usable datasets (fMRI+LSAS+behavior) for the imaging study. Effect sizes for individual differences (social anxiety) analyses will be imprecise and unstable.For example, the 95% CI's for a correlation of r=.35 in n=38 range from 0.035 (nil) to 0.602 (very strong).See also recent work by Schönbrodt and Perugini (2013) titled "At what sample size do correlations stabilize?".1b) This concern is amplified by the authors' use of a post-hoc median split collected from an unselected sample, which tends to further reduce power (decrease the precision of effect size estimates)1c) This concern is mitigated for the behavioral effects by the between-study replication (cf. Figure 2 panels C and D).

We agree with the reviewer that 38 participants is a small number for a study of interindividual differences. This was mainly due to technical limitations (e.g. access to our fMRI scanner) and the failure of 5 participants in holding their head still enough to allow effective motion correction (or rather our failure in installing participants comfortably in the scanner and reminding them of the importance of moving as little as possible during the scan). We now note in the Materials and methods (subsection “Participants”) that the number of participants recruited in Study 2 roughly corresponds to the sample size estimated for a point biserial model test based on the results of Study 1 [R^2^ = 0.13, one-tailed test with alpha error = 0.05 and power (1-beta) = 0.8, sample size = 43; G*Power 3.1]. Further, we originally excluded several participants due to their response pattern during the scanning experiment; in response to comments by reviewer 1, we have now included all subjects we could, bringing the total to 40, not a major improvement but at least a step in the right direction. We thank the reviewer for pointing us to the reference about stabilisation of correlations; this is now cited in the limitations paragraph and will guide planning of further studies.

Based on the review of the consequences of median splits proposed by the reviewer, we decided to completely remove median-split analyses from the manuscript and instead treat LSAS as a continuous variable throughout the manuscript. This change involved reanalysis of our data, leading to changes in all paragraphs of the Results section and in Figures 2 and 4, but did not substantially affect our behavioral or neuroimaging findings.

2) The value of the decoding analyses is unclear. Although the authors report that "decisions could indeed be decoded above chance from activation in the basolateral subregion of the right amygdala" the mean accuracy was 54.6% (chance = 50%), which provides very limited evidence. Moreover, it is unclear whether this cluster overlaps those moderated by social anxiety.

We agree that 54.6% prediction is very low. With the inclusion of additional participants after revision of our exclusion policy, prediction rose to 56.4%, but this value is still not far above chance. Further, we now provide overlap calculations in the Results (subsection “Neuroimaging results”), which shows that a part (29%) of the cluster identified in the decoding analysis lies within the cluster moderated by social anxiety, with 7.5% of the voxels of both clusters present in both of them. In light of these considerations, we considerably toned down our interpretation of the decoding findings, and now write that “Although decoding accuracy was relatively low, these findings are compatible with the involvement of a part of the amygdala in the decision-making process engaged by our task.”, and in the Discussion: “[…] while we could significantly decode participants’ choice from activation in the right amygdala, the average accuracy was quite low. Therefore, caution must be used in interpreting this finding: the amygdala cluster identified in our analysis is unlikely to be the major contributor to participants’ choices. Further studies specifically designed for a decoding analysis and investigating additional brain regions are required to better understand the neural mechanism underlying the decision-making process in our task.”

3) The imaging results and the behavioral results proceed in parallel and remain unconnected. Thus, claims that these results "suggest how activity differences in a subcortical network during social decision-making may lead to social avoidance" need to be tempered. Given the modest sample size, traditional statistical approaches for linking seem inadvisable.

We agree that we did not draw direct links between our behavioral results and the brain activation data in the original version of our manuscript – we only linked our experimental measure of social engagement to trait social anxiety, and brain activation to trait social anxiety. In this revised version, we now directly investigate the link between our measure of social engagement and brain activation, by using social engagement as a predictor in a linear regression attempting to explain the variation of BOLD responses in our regions of interest. We found that the left Amygdala response both during decision-making and when experiencing the outcome of game rounds decreased across participants with increasing value of social engagement (Figure 4C and 4E). We are of course aware that this link is not very strong given that this analysis is post-hoc, that the effects measured are not large and that the sample size was relatively low. However, this finding at least helps to connect the imaging and behavioral data a little better. These new findings are now reported in the Results section. We have further tempered the claims about the neural substrates of social avoidance that we make based on our findings, and now write “These findings suggest a relation between trait social anxiety/social avoidance and activity in a subcortical network during social decision-making.” (See the last sentence of the Abstract).

– gPPI and other functional connectivity measures do not license the interpretation of "information transmission".

We agree that the use of this term was not warranted. We replaced “information transmission” with “functional connectivity”, the term that we now use throughout the manuscript for these analyses.

– The use of a post-hoc median split is contrary to published recommendations from Kris Preacher and other statisticians.

We thank the reviewer for pointing us to this highly relevant literature. After reading Rucker, McShane and Preacher (2015), we better understand the issue with this approach and have now completely removed median splits from our manuscript.

– Interactions needs to be fully decomposed and adequately explained.

We now decomposed and explained in details the crucial interaction (neural response to risky vs. safe choice X human vs. computer partner) in this revised manuscript (subsection “Activation during the decision: Amygdala” and Figures 4A and 4B).

– "In other words, given an average earning in the game of 1.93 Euros (see Materials and methods), that person would lose 21.8% of the average earning in the game.""Our linear regression results suggest that persons likely to suffer from generalized social phobia (i.e. those with an LSAS score of 60) (21) would lose 21.8% of their earnings in this game."This is misleading because the average earnings reflect the average LSAS, not 0.

We apologize for this mistake, which we corrected by modifying our calculations reported as follows: “The average LSAS across participants of both studies was 29.02, their average earning 1.93 Euros, and the slope -0.008 Euros/LSAS point; therefore a person with an LSAS of 0 would earn 1.93+0.008*29 = 2.162 Euros or 12% more than the average participant, while a person with an LSAS of 60, likely to suffer from generalized social phobia (22), would earn 1.93-0.008*31 = 1.682 Euros or 12.8% less than the average participant and 22.2% less than a person with an LSAS of 0.”

Discussion/implications:– Fails to adequately acknowledge key study limitations.

We added a paragraph discussing in detail the limitations of our study (Discussion section, final paragraph), including the low number of participants and the uncertainty surrounding the estimated individual differences effects, the limited decoding accuracy and the inferences about social anxiety to be drawn from our particular set of participants with a likely restricted range of LSAS scores.

– The strength of the conclusions is not well calibrated to the strength of the evidence. The authors extrapolate a bit too optimistically based on a sample of 38 unselected young Germans, all of whom were willing to come to the lab and subject themselves to procedures that would be quite distressing to individuals with elevated levels of social anxiety (e.g. perform unfamiliar tasks in a novel setting while being scrutinized by unfamiliar experimenters).

We completely agree with the fact that our participant sample was very constrained and limited, and that our conclusions need to reflect this fact. Even those of our participants with high LSAS would clinically not be considered strongly socially anxious, as they would otherwise probably not have engaged in our experiments. We now describe these limitations in our extensive limitations paragraph, and tempered our conclusions in several places by referring to trait social anxiety (e.g. Abstract, last sentence and several additional places in the Results and Discussion sections).

Reviewer #3:This paper studies a risk task in which subjects choose whether to take a same amount of money or gamble to win more or get less. In the human condition upon win/loss they will see an admiring or condescending (positive/negative) face. The amount they will give up in expected earnings to avoid these facial 'judgments' is linked to social anxiety.There is a lot to like about this paper. There are two studies, the second replicating the first behaviorally (and is used for fMRI).One challenge is that there are four social anxiety scales used (LSAS and AQ in study 1, and two others added in study 2). One has to correct for multiple comparison in using all these scales as failing to do so inflates the apparent significance of the "best" scale

We agree that using several scales increases the likelihood of obtaining false positives. We now provide correction for the consequences of using multiple scales in a regression analysis with two recommended approaches, which we used in the analysis of the data of both studies. First, we assessed the overall significance of the complete model that includes all scales; this yielded only one global F value per study that takes into account the fact that there are several explanatory variables, including a reduction in degrees of freedom (these global F values were significant with *P*=0.028 in Study 1 and *P*=0.004 in Study 2; see subsection “Behaviour (study 1)” and “Behaviour (study 2, pre-fMRI)”). Second, we Bonferroni-corrected the P-values of the regressors selected in the backward stepwise regression, using the number of scales present in the full model (here, corrected values for LSAS, the regressor selected in both studies, were *P_corr_* = 0.012 in Study 1 and *P_corr_* = 0.010 in Study 2). These corrections thus relativize the significance of our results but do not invalidate our findings. The steps of the backward stepwise elimination are now reported in Supplementary file 1.

Admirable: study 1 and 2 replicate behavioral pattern. This is really great.I like the use of an economic measure of the "cost" of social anxiety. As authors note at the tail end of the paper, one could do a lot of interesting pricing of disorders in this way. There probably are similar studies in economics, but often use inferior measures of disorder and of course it is hard to capture all the costs and benefits. In any case, it is great to see this calculation put forward here even though it is not ideal.

We thank the reviewer for this encouraging comment (the robust effect observed in Study 1 prompted us to invest in a follow-up neuroimaging study) and agree that a combination of experimental and economic measures to better assess the impact of social anxiety on a person’s life would be optimal. We probably just scratched the surface of what is possible in this regard and are looking forward to developing further related measures or approaches to assess costs of disorders and personality traits.

Not sure why the term "opponents" are used. Do these facial opponents gain/lose when the subjects lose/win?

We agree that “opponent” was not a good term (the opponents did not gain or lose anything) and replaced it throughout the text with “partner”.